# Effect of Sphingomyelinase-Treated LDLs on HUVECs

**DOI:** 10.3390/molecules28052100

**Published:** 2023-02-23

**Authors:** Angelica Giuliani, Camilla Morresi, Gabriele Mazzuferi, Luisa Bellachioma, Deborah Ramini, Jacopo Sabbatinelli, Fabiola Olivieri, Tiziana Bacchetti, Gianna Ferretti

**Affiliations:** 1Department of Clinical and Molecular Sciences, Polytechnic University of Marche, 60126 Ancona, Italy; 2Department of Life and Environmental Science, Polytechnic University of Marche, 60100 Ancona, Italy; 3Department of Clinical Science, Polytechnic University of Marche, 60100 Ancona, Italy; 4Clinic of Laboratory and Precision Medicine, IRCCS INRCA, 60121 Ancona, Italy; 5Laboratory Medicine Unit, Azienda Ospedaliero Universitaria delle Marche, 60126 Ancona, Italy

**Keywords:** sphingomyelinase, ceramide, low-density lipoprotein, oxidative stress

## Abstract

Low-density lipoproteins (LDLs) exert a key role in the transport of esterified cholesterol to tissues. Among the atherogenic modifications of LDLs, the oxidative modification has been mainly investigated as a major risk factor for accelerating atherogenesis. Since LDL sphingolipids are also emerging as important regulators of the atherogenic process, increasing attention is devoted to the effects of sphingomyelinase (SMase) on LDL structural and atherogenic properties. The aims of the study were to investigate the effect of SMase treatment on the physical-chemical properties of LDLs. Moreover, we evaluated cell viability, apoptosis, and oxidative and inflammatory status in human umbilical vein endothelial cells (HUVECs) treated with either ox-LDLs or SMase-treated LDLs (SMase-LDLs). Both treatments were associated with the accrual of the intracellular ROS and upregulation of the antioxidant Paraoxonase 2 (PON2), while only SMase-LDLs induced an increase of superoxide dismutase 2 (SOD2), suggesting the activation of a feedback loop to restrain the detrimental effects of ROS. The increased caspase-3 activity and reduced viability observed in cells treated with SMase-LDLs and ox-LDLs suggest a pro-apoptotic effect of these modified lipoproteins on endothelial cells. Moreover, a strong proinflammatory effect of SMase-LDLs compared to ox-LDLs was confirmed by an increased activation of NF-κB and consequent increased expression of its downstream cytokines IL-8 and IL-6 in HUVECs.

## 1. Introduction

Low-density lipoproteins (LDLs) exert a key role in transport of esterified cholesterol to tissues. An elevated plasma concentration of cholesterol associated with LDLs (LDL-C) is a primary causal factor in the development of atherosclerotic cardiovascular disease and significantly contributes to the cardiovascular risk [1,2]. The LDL surface contains several apoproteins and amphipathic lipids. Sphingomyelin and phosphatidylcholine are the main phospholipids, and the LDL hydrophobic core contains cholesterol esters and triglycerides [3]. Previous studies have shown that lipoprotein lipids exert a conformational role on apoproteins. In this regard, alterations of the interactions between lipids and apoprotein B100, together with lipid peroxidation, can contribute to functional alterations of LDLs [4,5]. Among the atherogenic modifications of LDLs, the oxidative modification has been mainly investigated as a major risk factor for accelerating atherogenesis [6]. In fact, several studies have demonstrated that oxidized LDLs (ox-LDLs), regardless of the prooxidant agent or stimulus, display alterations of their physical-chemical properties and altered interactions with cell receptors [4,7]. In macrophages, an enhanced uptake of ox-LDLs by scavenger receptors can lead to accumulation of cholesterol ester and formation of foam cells. Using human umbilical vein endothelial cells (HUVECs), it has been demonstrated that ox-LDLs induce the expression of adhesion molecules on the cell surface, an early event in atherogenesis [4,6]. During their life in circulation, LDLs are susceptible to several compositional changes due to lipid transfer proteins, enzymes of lipoprotein metabolism and cell enzymes. Lipolytic modifications of LDLs triggered by cell enzymes such as the group V secretory phospholipaseA2 (PLA2)–secreted by macrophages [8] and by sphingomyelinase (SMase) released by macrophages and endothelial cells, contribute to a higher atherogenicity [9,10].

Among the lipolytic changes of LDLs, an increasing attention is devoted to the key role attributed to the effects of SMase on LDL structural and atherogenic properties [11,12]. SMase is a sphingomyelin-specific form of phospholipase C that catalyze the cleavage of the phosphodiester bond from sphingomyelin, resulting in the production of ceramide and phosphocholine. LDL-derived sphingolipids are strongly implicated in several pathologic conditions, including insulin resistance and cardiovascular events. For instance, recent studies have demonstrated that the susceptibility of LDL particles to aggregate depends on their sphingomyelin content, and that the latter is associated with future cardiovascular death [13]. One mechanism for the LDL aggregation of SMase-treated LDLs focuses on the formation of hydrophobic domains on the surface of the LDL phospholipid monolayer. This occurs due to SMase-catalyzed hydrolysis of sphingomyelin, which accounts for ∼20–25% of the total LDL phospholipids [14]. A pathophysiological role of the SMase activity is supported by data which demonstrate that this enzyme is secreted by macrophages and endothelial cells in the vicinity of typical atherosclerotic lesional tissue [15]. Therefore, the hydrolysis of LDL-sphingomyelin by secretory SMase is hypothesized to contribute to the formation of atherosclerotic tissue [15]. SMase treatment of LDL promotes the uptake of lipoproteins and accumulation of cholesterol within macrophages [16]. Inflammation plays a crucial role in all phases of the atherosclerotic process, involving both endothelial and immune cells. IL-6 levels were associated with an increased cardiovascular risk, since IL-6 can affect different type of cells involved in lipid processing and plaque formation [17,18].

The aims of the present study were to investigate the effect of SMase treatment on the physical-chemical properties of LDLs and compare their characteristics and biological activities with the most extensively investigated LDL modification, i.e., oxidation. 

Moreover, to gain further insights into the atherogenic potential of SMase-treated LDLs, we compared the effects of ox-LDLs and SMase-treated LDLs on an endothelial cell model (HUVECs) by evaluating markers of cell viability, apoptosis, pro-inflammatory status, oxidative stress, and antioxidant defense.

## 2. Results

### 2.1. Sphingomyelinase Treatment Induces Physico-Chemical Modification in LDL

Table 1 summarizes the physico-chemical properties of control LDLs, ox-LDLs and SMase-LDLs. The study of lipid peroxidation has shown a significant increase of levels of Thiobarbituric acid reactive substances (TBARS) only in ox-LDLs. The level of ceramide was significantly increased in SMase-treated LDLs, and the level was slightly increased even in ox-LDLs. Apo B100 structural properties in control LDLs and treated LDLs were investigated using the intrinsic fluorescence of the tryptophan residues (Trp) of the apoprotein. A significant decrease of fluorescence intensity was observed in ox-LDLs. Treatment of LDLs with SMase for 24 h also induced a decrease in the fluorescence intensity of the Trp residues, but to a lower extent compared with ox-LDLs (Table 1). The decrease in fluorescence intensity was significant at the highest enzyme concentration (100 mU/mL). These data demonstrate that changes in the lipid components are reflected by conformational changes of the ApoB100 associated with LDLs. Changes in the apoprotein component of SMase-LDL were also investigated by evaluating hyperchromicity at 282 nm. LDL samples showed an increase in absorbance at 282 nm following treatment with SMase. The increase was significant at the highest enzyme concentration (100 mU/mL) (Table 1).

Using the fluorescence emission spectra of the probe Laurdan, we also observed modifications of the LDL physico-chemical properties with a decrease of the value of generalized polarization (GP) in LDL samples incubated with SMase at the higher concentration (GP = 0.42 ± 0.02 in SMase-LDL), compared with LDLs incubated without the enzyme (C-LDL, GP = 0.52 ± 0.01). The analysis of absorbance at 450 nm of LDL samples treated with SMase has shown a significant increase in turbidity, in comparison with untreated LDLs (LDL-C: 0.291 ± 0.012; LDL + SMase 50 mU/mL: 0.545 ± 0.021 AU; LDL + SMase 100 mU/mL: 0.797 ± 0.025 AU). No significant changes of turbidity were observed in ox-LDLs.

### 2.2. Modified LDLs Decreased Cell Viability and Increased Caspase-3 Expression in HUVECs

Figure 1 shows the viability of HUVECs, detected by the 3-(4,5-dimethylthiazol-2-yl)-2,5-diphenyltetrazolium bromide (MTT) assay after 24 h of incubation with LDLs treated in different experimental conditions. Viability was not significantly modified in cells incubated with control LDLs with respect to cells incubated without lipoproteins. A decrease of viability was observed in cells incubated with modified LDLs compared to cells incubated in the presence of control LDLs (*p* < 0.05). The effect on cell viability was dependent on the LDL concentration. The decrease in cell viability was seen to a greater extent using the highest concentration of ox-LDL (50 µg/mL). A significant decrease was also observed after incubation for 24 h with LDLs treated with the highest concentration of SMase (100 mU/mL).

To evaluate a potential activation of the proteins involved in apoptosis, the expression of the caspase-3 protein in HUVECs was carried out following incubation with normal and modified LDLs. A significant increase of caspase-3 expression was observed in cells incubated with ox-LDLs and with SMase-LDL 100 mU/mL (Figure 2).

### 2.3. Modified LDLs Modulate Oxidative Status and Antioxidant Defense of HUVECs

In order to study whether ox-LDLs and SMase-LDLs are able to induce alterations of the oxidative status of endothelial cells, the intracellular levels of reactive oxygen species (ROS) were compared in HUVECs incubated in different experimental conditions. Incubation of HUVECs for 24 h with 20 µg/mL LDL had no significant effects on the intracellular levels of ROS assessed using the 2′,7′-dichlorodihydrofluorescein diacetate (DCFH_2_-DA, DCF) probe compared to untreated cells. A higher concentration of 50 µg/mL LDL caused a significant increase in DCF fluorescence (about 60%, *p* < 0.05) (Figure 3). Similar to ox-LDLs, a significant increase of ROS was also observed in cells incubated with SMase-LDLs (*p* < 0.05; Figure 3).

The expression of the antioxidant enzyme Paraoxonase-2 (PON2) was significantly increased in HUVECs incubated with ox-LDLs and SMase-treated LDLs, while the expression of the antioxidant enzyme Superoxide dismutase (SOD 2) was upregulated only after treatment with SMase-LDLs. Our findings suggest that endothelial cells are likely to initiate a response against oxidative stress (Figure 4).

### 2.4. Ox-LDLs and SMase-LDLs Induce a Proinflammatory Response in HUVECs

To further study the molecular mechanisms that can contribute to the alterations of cell viability and to the increase of ROS in HUVECs in response to incubation with modified LDLs, we analyzed the expression of the transcription factor NF-κB and of its downstream proinflammatory cytokines IL-8 and IL-6. The expression of the phosphorylated p65 subunit, which reflects NF-κB activation, was significantly increased in HUVECs in the presence of ox-LDLs and SMase-LDLs at a concentration of 50 µg/mL (Figure 5).

An increased expression of IL-6 and IL-8 was observed in HUVECs treated with either ox-LDLs or SMase-LDLs, with the latter having a greater effect (Figure 6). Accordingly, IL-6 secretion in cellular media confirmed a greater proinflammatory action of SMase-LDL compared to ox-LDL on HUVECs.

## 3. Discussion

The lipid peroxidation of LDLs occurs in vivo and is triggered by different factors [19,20]. The lipid composition changes and the atherogenic properties of ox-LDLs have been previously studied using different cell models [21,22,23], and the effects have been extensively reviewed [24,25]. The uptake of oxidized low-density lipoprotein through scavenger receptors by endothelial cells drives the activation of transcription factors, such as nuclear factor-κB (NF-κB), that evoke proinflammatory adhesion molecule expression in endothelial cells [23]. Among the atherogenic alterations, an increasing attention is devoted to the effects of sphingomyelinase (SMase) released by macrophages and endothelial cells on LDLs [9]. In fact, the plasma levels of sphingomyelin, upon enzymatic hydrolysis by sphingomyelinase yields ceramide, have been shown to correlate with the severity of coronary artery disease [13]. The increase in LDL ceramide levels is considered as a key factor contributing to the aggregation of LDLs within the arterial wall. In fact, biophysical studies have shown that ceramide has a pronounced tendency to self-aggregation. Therefore, the formation of ceramide from sphingomyelin is considered a critical step in atherosclerosis [26]. In our experimental conditions, alterations of the lipid composition and physico-chemical properties of LDLs induced by treatment with SMase have been confirmed by a significant increase in ceramide level compared with untreated LDLs, which is in good agreement with other studies [27]. In addition, we demonstrated an increase of turbidity and a decrease of molecular order on the phospholipid surface of the LDLs in the microenvironment of the probe Laurdan. The effect was dependent on the concentration of SMase and was significant at 100 mU/mL SMase. LDLs treated with SMase also showed alterations of ApoB100 as shown by the decrease in Tryptophan fluorescence and by the increase of hyperchromicity at 282 nm. All these results suggest alterations of ApoB100 structure and a greater exposure of ApoB100 aromatic amino acid residues in LDLs treated by SMase, which is in good agreement with other authors [28].

We confirmed a cytotoxic effect exerted by ox-LDLs on HUVECs, with higher ROS levels and activation of NF-κB in our experimental conditions, which is in agreement with Cominacini et al. [23]. Previous studies have shown that ox-LDLs cause an increase of ROS production in different cell models, including fibroblasts and endothelial cells [29,30]. It has been suggested that mitochondria could be a possible source of ROS in endothelial cells following incubation with ox-LDLs. It was also shown that ox-LDLs can activate nitric oxide synthase (eNOS), leading to the activation of c-Jun NH2-terminal kinase (JNK) [31]. In addition, ROS have been proposed to selectively activate phosphorylation of NF-κB via a redox-regulated tyrosine kinase [32].

The effects of SMase-LDLs on HUVEC have not been previously studied. We confirmed that in our experimental conditions, aggregation occurs at the end of incubation between LDLs and SMase. The literature data demonstrate that aggregated LDLs can be internalized by an LDL receptor or other mechanisms involving plasma membrane invaginations. Previous studies have demonstrated uptake of SMase-LDLs mediated by LDL receptors in macrophages [16,33]. In addition, using cytochalasin D during incubations, it has been demonstrated that endocytosis, not phagocytosis, was involved in the internalization of SMase-treated LDLs. Boyanovsky et al. [34] have confirmed that the uptake of ceramide-enriched LDLs by human microvascular endothelial cells in a receptor-mediated fashion. Further studies are necessary to investigate the molecular mechanisms of the interactions between SMase-LDLs and HUVEC. However, we demonstrated that SMase-LDLs exert a cytotoxic effect on HUVEC, cause a significant increase of ROS, and stimulate NF-κB activation, as well as the production of IL-6 and IL-8, with respect to unmodified LDLs. Importantly, IL-6 secretion in cellular media confirmed a greater proinflammatory action of SMase-LDLs compared to ox-LDLs on endothelial cells. IL-6 is a pivotal cytokine of innate immunity, modulating a broad set of physiological functions traditionally associated with host defense, immune cell regulation, proliferation, and differentiation [35]. Extensive literature supports the proatherogenic role for IL-6 in cardiovascular disease, and IL-6 inhibition has consequently been proposed as a novel method for vascular protection [36]. Therefore, we can hypothesize that interactions between SMase-LDLs and the plasma membrane of HUVECs could activate cytotoxic mechanisms, increase ROS, and trigger proinflammatory responses mediated by the NF-κB pathway. In addition, the increased expression of IL-8, the most relevant chemokine in the framework of in vivo inflammation, strongly suggests a proinflammatory activity of SMase-LDLs. The effect of SMase-LDLs could be related to alterations of ApoB100 structure and/or to the increase of ceramide levels. Our hypothesis is supported by literature data. A cytotoxic effect was observed from LDL(−) particles isolated from animal models [37]. As previously mentioned, LDL(−) have a higher level of ceramide and show structural alterations of the apoB100 [14]. A potential role in inflammatory signaling exerted by SMase-LDLs has been observed in monocytes with an SMase-induced monocyte arachidonic acid release and cPLA_2_ activation, accompanied by increased TNF-α secretion [38]

We also demonstrated a significant increase of expression of caspase-3 in HUVECs treated with ox-LDLs or SMase-LDLs. The activation of caspases has been previously observed in human coronary artery endothelial cells (HCAEC) incubated with ox-LDLs [39]. Caspase activation can damage the permeability and integrity of the mitochondrial membrane, decreasing the mitochondrial membrane potential and destroying the structure of the mitochondrial membrane. Some hypotheses can be formulated to explain the effect of SMase-LDLs on caspases. We hypothesize that the interactions between SMase-LDLs and HUVECs result in the activation of the expression of the caspase and the potential activation of apoptosis. The effect could be attributed to the higher level of ceramide, as hypothesized by Pettus et al. [40]. Other authors have shown that ceramide-rich LDLs are taken up by receptor-mediated mechanisms and can deliver excess ceramide to the cells [34]. The accumulation of LDL-derived ceramide within cells induces apoptosis of HME-1 cells [34]. Ceramide has also been shown to activate reactive oxygen species (ROS), mitochondrial oxidative damage, and apoptosis in vascular cells [41].

PON2 is an antioxidant intracellular enzyme localized in mitochondria, endoplasmic reticulum and in plasma membrane, and is expressed in several cells [42,43,44]. PON2 exerts a protective effect against oxidative damage triggered under different experimental conditions [42,43,44]. A significant increase in PON2 expression was observed in cells incubated with ox-LDLs or SMase-LDLs. An increased expression of PON2 has been previously observed in HUVEC in response to oxidative stress caused by glycation end products such as glycated albumin (GA) and Nε-(carboxymethyl) lysine (CML) [45]. We suggest that the higher expression of PON2 could represent a defense mechanism activated by the cell in response to higher ROS levels. This hypothesis is supported by previous studies on Caco-2 cells with a higher PON2 expression in response to inflammatory agents and to higher expression of NF-κB [46], and further supported by the increased expression of the antioxidant enzyme SOD2.

In conclusion, although ox-LDL and SMase-treated LDLs show different compositional and physico-chemical properties, they trigger ROS increase in HUVECs and increase phosphorylation of NF-κB. We suggest that SMase-LDLs could trigger free radical signaling through activation of one or more of the many enzymatic sources for ROS which are present in almost all cell types and that these free radicals may play a key role in the cellular pathways leading to NF-κB expression. 

Our results could have a physiological relevance. Beyond their structural role, it is now clearly established that sphingolipids serve as bioactive signaling molecules to regulate diverse processes including inflammatory signaling and proliferation. An increase of SMase activity has been demonstrated in the plasma of patients affected by dysmetabolic diseases [47,48]. Secretory SMase activity is increased in the serum of patients with type 2 diabetes, chronic heart failure, and acute coronary syndromes [47,48]. In addition to the Mg^2+^-dependent SMase, a Zn^2+^-dependent SMase is secreted by macrophages and endothelial cells in the vicinity of typical atherosclerotic lesional tissue [49]. Tissue containing LDLs from atherosclerotic lesions contains 10–50 times more ceramide than intact plasma LDLs within lesional tissue [10,15]. In patients with metabolic syndrome or diabetes, circulating plasma ceramide levels are significantly higher than in normal individuals. Electronegative LDLs show SMase activity, which leads to increased ceramide levels that can produce pro-inflammatory effects and susceptibility to aggregation [50].

## 4. Conclusions

In conclusion, although ox-LDLs and SMase-treated LDLs show different compositional and physico-chemical properties, they trigger ROS increase in HUVEC and increase the phosphorylation of NF-κB. We suggest that alterations of the interactions between SMase-LDLs and cell plasma membrane could trigger free radical signaling through activation of one or more of the many ubiquitarian enzymatic sources of ROS and that these free radicals may play a key role in the cellular pathways leading to NF-κB expression. Moreover, in our experimental conditions SMase-LDLs exert a proinflammatory effect that has not been previously shown.

## 5. Materials and Methods

LDLs from human plasma were purchased from Prospec-TanyTechnoGene Ltd. (Ness-Ziona, Israel). Copper sulphate, methylglyoxal, Hepes, PBS, EDTA, and NaCland sphingomyelinase (SMase) from *B. cereus* were acquired from Merk Life Science S.r.l.(Milan, Italy). 6-Dodecanoyl-2-Dimethylaminonaphthalene (Laurdan) probe was purchased from Thermo Fisher Scientific (Waltham, MA, USA); Bradford reagent was purchased from Bio-Rad Laboratories S.r.l. (Segrate (MI), Italy). Amplex Red kit was purchased from Thermo Fisher Scientific (Waltham, MA USA). HUVECs (human umbilical vein endothelial cells) were purchased from Clonetics Corporation (Lonza, Basel, Switzerland) and cultured in a special growth medium for endothelial cells (Endothelial Growth Medium-2-EGM, by Lonza, Basel, Switzerland). Caspase-3 (CAS3), nuclear factorkB p65 phosphorylated (Fosfo-NF-κB p65), actin, vinculin, and GADPH antibodies were acquired from Euroclone (Pero (MI), Italy), while Paraoxonase 2 (PON2) antibody was from Merk Life Science S.r.l. (Milan, Italy).

### 5.1. Modifications of LDLs

#### 5.1.1. Oxidation of LDLs

LDL (1 mg/mL) was oxidized by incubation for 24 h at 37 °C with 10 μM copper sulphate according to the literature [51]. At the end of incubation, the oxidation was stopped by addition of EDTA (final concentration: 10 mM). Treated LDLs were dialysated in PBS at pH 7.4, for 24 h at 4 °C.

#### 5.1.2. Treatment of LDL Sphingomyelinase

LDLs were treated with SMase from *Bacillus cereus*. Incubations were conducted using LDLs (1 mg/mL) in 5 mMHepes, 150 mMNaCl for 24 h at 37 °C. Two concentrations of SMase (50 mU/mL and 100 mU/mL) were used [52]. At the end of incubation, the lipolysis was stopped by addition of EDTA (final concentration: 10 mM). The hydrolysis of sphingomyelin was followed using the Amplex Red-phosphorylcholine-coupled SMase fluorescence assay kit. LDL samples were treated with1 unit/mL horseradish peroxidase, 0.1 unit/mL choline oxidase, 4 units/mL alkaline phosphatase, and 50.5 μM Amplex Red Reagent in HEPES buffer. The progress of the sphingomyelin hydrolysis was observed as an increase of resorufin fluorescence using excitation and emission wavelengths, 530–569 nm and 590 nm, respectively. As resorufin is produced in equimolar amounts with the Phosphocholine and ceramide, the fluorescence is proportional to ceramide generation.

A calibration curve was made using increasing concentrations of phosphorylcholine in HEPES buffer in order to measure ceramide formation in LDL treated in the absence or in the presence of SMase [52].

### 5.2. Measurement of LDL Peroxidation

The formation of thiobarbituric acid (TBA) products were evaluated in LDLs treated in different experimental conditions using absorbance at 535 nm [53]. 1,1,3,3-Tetramethoxypropane was used as standard. All measurements were conducted in triplicate. Data are presented as mean ± SD.

### 5.3. Measurement of LDL Aggregation

Measurement of LDL aggregation was analyzed monitoring turbidity at an absorbance of 450 nm. All measurements were conducted in triplicate [28].

### 5.4. Measurement of LDL Hyperchromicity

Hyperchromicity of LDLs treated in the different experimental conditions was evaluated at 280  nm. The hyperchromicity of sample at 280  nm reflects the exposure of chromophoric aromatic amino acid residues due to the unfolding and fragmentation of Apo B [19].

### 5.5. Physico-Chemical Properties of Untreated and Modified LDLs

Laurdan probe was dissolved in a 100% methanol solution (concentration 1 mM) and stored at −20 °C. Briefly, an aliquot of Laurdan was incorporated with LDL (100 μg/mL of protein) for 30 min at 37 °C, using a final probe concentration of 1 μM. [51].

From the emission spectra obtained, the intensities at 440 nm and 490 nm were considered for the calculation of the GP (generalized polarization) parameter through the equation:GP = (I 440 − I 490)/(I 440 + I 490). A high GP value is associated with a lower membrane fluidity and a lower polarity of the microenvironment surrounding the probe [54]. All measurements were conducted in triplicate. 

### 5.6. Intrinsic Fluorescence Spectroscopy 

The intrinsic fluorescence resulting from the presence of aromatic amino acids (tryptophan and tyrosine) of Apo-B100 was used to study apoprotein properties using emission spectra (excitation wavelength = 295 nm). The position of the maximum emission of tryptophan fluorescence is sensitive to the hydrophobicity of the surrounding environment. The samples of untreated LDLs and modified LDLs were resuspended in 5 mM Hepes buffer, 150 mM NaCl at pH 7.4. The emissions spectra were evaluated using a Perkin Elmer LS 55 spectrofluorimeter. All measurements were conducted in triplicate [51].

### 5.7. Cell Culture 

Cryopreserved HUVECs obtained from a pool of three donors were purchased from Clonetics (CC-2519, Lonza, Basel, Switzerland) and maintained in EBM-2 (CC-3156, Lonza) supplemented with SingleQuot Bullet Kit (CC-4176, Lonza) and maintained in a humidified atmosphere of 5% CO2 at 37 °C. Cells were seeded in flasks up to passage five at a density of 5000/cm^2^ and sub-cultured when they reached 70–80% confluence. All cell cultures were regularly tested for mycoplasma contamination.

8 × 10^5^ HUVECs were cultured in 6-well plates and allowed to attach overnight before the treatment. Then, cells were exposed to buffer (lipoprotein free, negative control) or incubated with untreated LDLs, oxidized LDLs (ox-LDLs), or LDLs treated with sphingomyelinase (SMase-LDLs). Before incubation with cells, all LDL samples were passed in 0.2 µm filters under sterile conditions. Cells were treated in absence or in the presence of control or modified LDLs (20 µg/mL and 50 µg/mL) for 24 h; cells were then harvested and used for subsequent analysis [55].

### 5.8. Cell Viability Assay

Cell viability was determined using the colorimetric assay based on the reduction of a yellow tetrazolium salt (3-(4,5-dimethylthiazol-2-yl)-2,5-diphenyltetrazolium or MTT) to purple formazan crystals by metabolically active cells. HUVECs were cultured into 96 well plates at density of 8 × 10^3^ cells/cm^2^ [56]. After 24 h, the cells were washed with fresh medium and then treated in absence of lipoproteins, or with normal LDLs, ox-LDLs, and SMase-LDLs. After 24 h of treatment, the MTT solution (1 mg/mL) was added and incubated for an additional 4 h. The obtained product was solubilized with 200 μL of dimethyl sulfoxide (DMSO). Absorbance was measured by a microplate reader (MPT Reader, Invitrogen, Milano, Italy) at the optical density of 540 nm. All measurements were conducted in triplicate.

### 5.9. Intracellular ROS Levels

ROS levels were analyzed using 2′,7′-dichlorodihydrofluorescein diacetate (DCFH2-DA) probe (Sigma-Aldrich, St. Louis, MO, USA). Upon reaching p4, the cells were passed into 96 well plates (density 2 × 104 cells/mL), with a volume of 100 μL of medium for each well. After 24 h, the cells were washed with fresh medium and then treated without lipoproteins or incubated with normal LDLs, oxidized LDLs, or SMase-treated LDLs using LDL concentrations of 20 μg/mL and 50 μg/mL. After 24-h incubation, the medium was removed, and after washing with PBS, the samples were incubated in the dark at 37 °C with 25 μM DCFH_2_-DA for 30 min. Cells were washed to remove extracellular DCFH_2_-DA and then phosphate-buffered saline was added. The fluorescence of the cells from each well was measured and recorded on a fluorescence plate reader at λex/λem (485/535 nm) (Multi-Mode Microplate Reader SynergyTM HT, Agilent Technologies Italia S.p.A. Cernusco sul Naviglio (MI), Italy) [57]. All measurements were conducted in triplicate.

### 5.10. Western Blot

RIPA buffer (150 mMNaCl, 10 mM Tris, pH 7.2, 0.1% SDS, 1.0% Triton X-100, 5 mM EDTA, pH 8.0) containing a protease inhibitor cocktail (Roche Applied Science, USA) and a phosphatase (Sigma-Aldrich) inhibitor cocktail was used to obtain the cell lysates. Protein concentration was determined using Bradford Reagent (Sigma-Aldrich, Milano, Italy). Total protein extracts (25 μg) were separated by SDS-PAGE and then transferred to a nitrocellulose membrane using the Trans-Blot Turbo™ Transfer system (Bio-Rad). Membranes were then blocked for 1 h at room temperature (RT) in TBS with 0.1% of Tween-20 containing 5% non-fat dried milk, and subsequently incubated overnight at 4 °C with the primary antibodies of interest. All primary antibodies were probed with a secondary horseradish peroxidase (HRP)-conjugated antibody (Vector, USA). Proteins were visualized by ECL and the chemiluminescent signaling acquired using ChemiDoc XRS + System (Bio-RadLaboratories, Hercules, CA, USA) and analyzed using Image J software (Version 1.50i, National Institute of Health, Bethesda, MD, USA).

The primary antibodies used were Caspase-3 (CAS3) (# 9664) (Cell Signaling Tecnologies), Phospho-NF-κB p65 (# 5970), NF-κB p65 (Sc-8008) (Santa Cruz Biotechnologies) Paraoxonase 2 (PON2) (# SAB2700275). Actin, vinculin, and glyceraldehyde-3-phosphate dehydrogenase (GADPH) were used as normalizers.

### 5.11. ELISA Assay

Culture medium from the cell cultures was collected at the end of each incubation, centrifuged at 14.000 × RPM for 20 min, and stored at −80 °C until use. IL-6 concentration was measured using a commercially available enzyme-linked immunosorbent assay (ELISA) kit. IL-6 concentration was determined in triplicate according to the instructions from the manufacturers (#501030, Cayman Chemical Ann Arbor, MI, USA).

### 5.12. RNA Isolation and MRNA Expression

Total RNA was isolated using the NorgenBiotek Kit (#37500, Thorold, ON, Canada), according to the manufacturer’s instructions. RNA was stored at −80 °C until use. RNA amount was determined by spectrophotometric quantification with Nanodrop ONE (NanoDrop Technologies, Wilmington, NC, USA). Total RNA (1000 ng) was reverse-transcribed using TAKARA Kit (PrimeScript™ RT reagent Kit with gDNA Eraser, Cat: RR047A) based on the manufacturer’s instructions. RT-PCR was performed in a Rotor-Gene Q (Qiagen, Hilden, Germany) using TB Green™ Premix Ex Taq™ (Cat: RR420A) in a 10 µL reaction volume. mRNA quantification was assessed using the 2^−ΔCT^ method. β-actin were used as an endogenous control. Each reaction was run in duplicate and always included a no-template control.

The primers’ sequences (written 5′-3′) were: IL-6, Fw: CCAGCTACGAATCTCCGACC, Rv: CATGGCCACAACAATGACG; IL-8, Fw: TCTGCAGCTCTGTGTGTGAAGG, Rv: TGGGGTGGAAAGGTTTGGA; β-actin, Fw: TGCTATCCCTGTACGCCTCT, Rv: GTGGTGGTGAAGCTGTAGCC; SOD2, Fw: GTT GGG GTT GGC TTG GTT TC, Rv: ATA AGG CCT GTT GTT CCT TGC.

### 5.13. Statistical Analysis

Data are presented as mean ± standard deviation (SD) of at least 3 independent experiments. Two-tailed paired Student’s t test was applied to determine differences between samples. *p* values < 0.05 were considered significant.

## Figures and Tables

**Figure 1 molecules-28-02100-f001:**
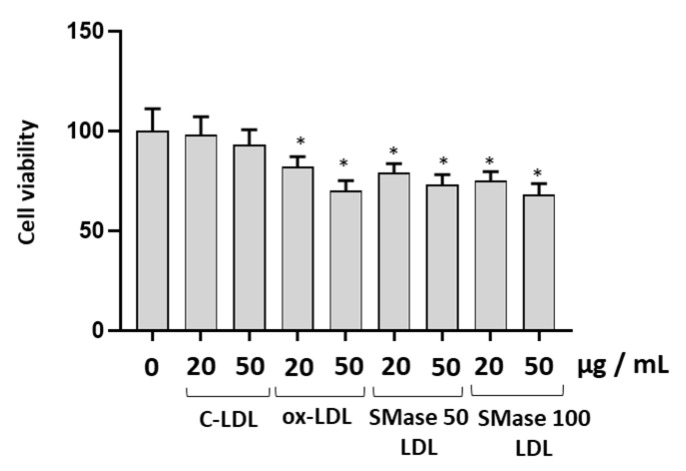
The effect of modified LDLs on cell viability in HUVECs. Cell viability was evaluated in HUVEC cells incubated for 24 h in the absence or presence of control LDLs (C-LDLs), oxidized LDLs (ox-LDLs) and LDLs treated with 50 mU/mL or 100 mU/mL SMase (SMase-LDLs). Results are presented as mean ± SD of 3 determinations carried out in triplicate. (* *p* < 0.05 vs. cells incubated with C-LDL).

**Figure 2 molecules-28-02100-f002:**
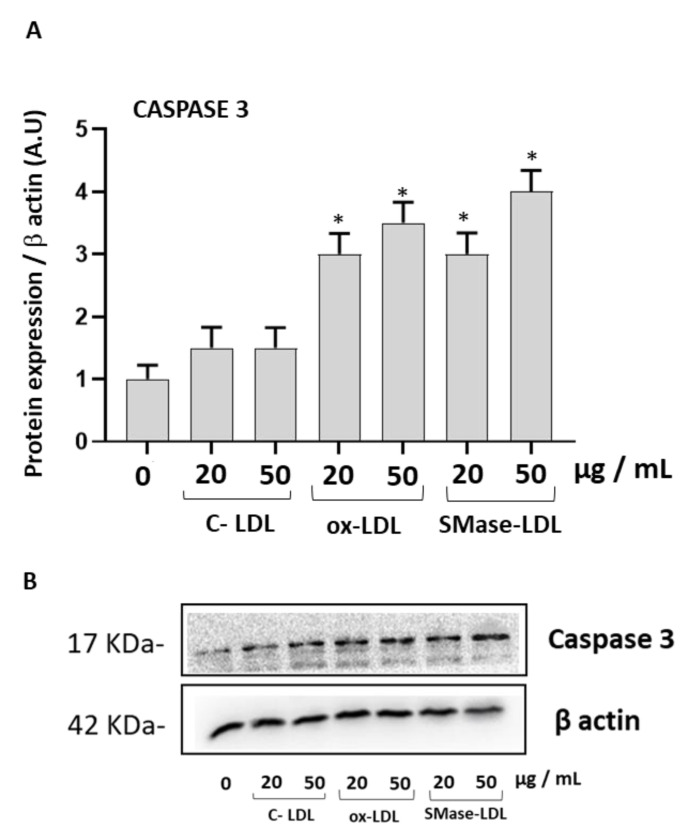
The effect of modified LDLs on Caspase-3 in HUVECs. (**A**) Densitometric analysis of Caspase-3 in HUVEC cells incubated for 24 h in the absence or in the presence of control LDLs (C-LDLs), oxidized LDL (ox-LDLs) and LDLs treated with 100 mU/mL SMase (SMase-LDLs). Data are normalized on β actin. Results are presented as mean ± SD of 3 determinations carried out in triplicate. (* *p* < 0.05 vs. cells incubated with C-LDL), (**B**) representative western blot images of caspase-3.

**Figure 3 molecules-28-02100-f003:**
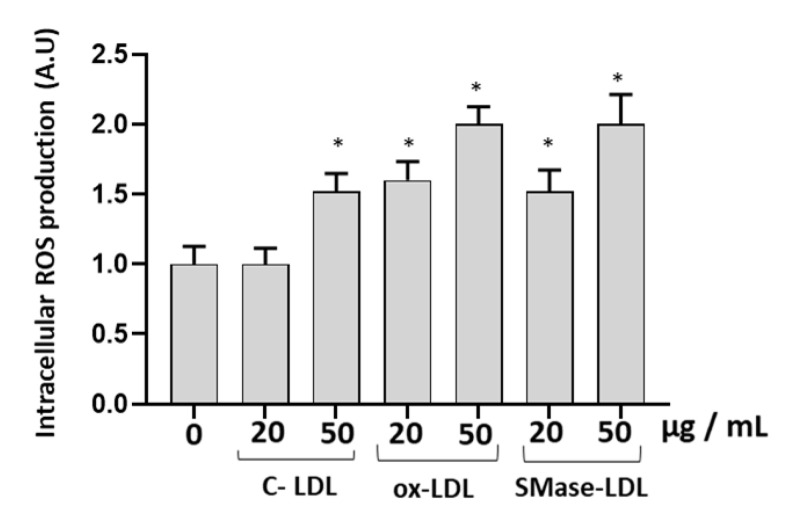
The effect of modified LDLs on intracellular ROS production in HUVECs. Intracellular ROS production in HUVECs incubated for 24 h in the absence or in the presence of control LDLs (C-LDLs), oxidized LDLs (ox-LDLs) and LDLs treated with 100 mU/mL SMase (SMase-LDLs). Results are presented as mean ± SD of 3 determinations carried out in triplicate. (* *p* < 0.05 vs. cells incubated with C-LDL).

**Figure 4 molecules-28-02100-f004:**
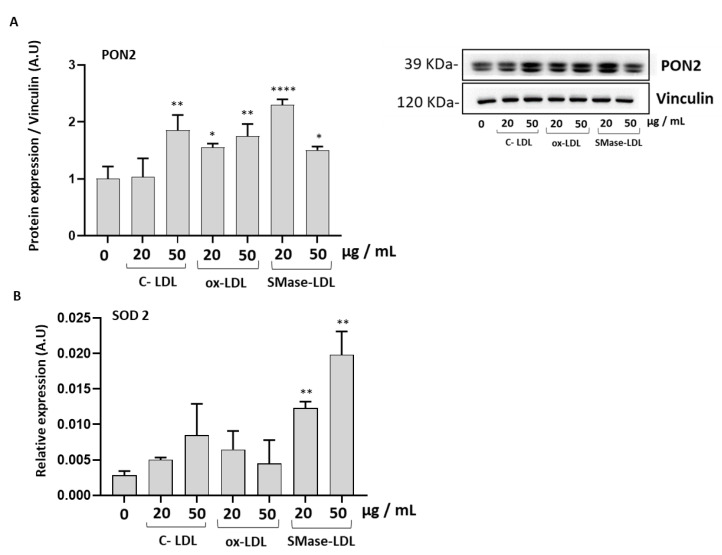
The effect of modified LDLs on levels of PON2 and SOD2 in HUVECs. (**A**) Densitometric analysis and representative western blot images of PON2; (**B**) relative mRNA levels of SOD2 in HUVEC cells incubated for 24 h in the absence or in the presence of control LDLs (C-LDLs), oxidized LDLs (ox-LDLs) and LDLs treated with 100 mU/mL SMase (SMase-LDLs). Results are presented as mean ± SD of 3 determinations carried out in triplicate. Densitometric data are normalized on Vinculin. (* *p* < 0.05, ** *p* < 0.001, **** *p* < 0.0001 vs. cells incubated with C-LDL).

**Figure 5 molecules-28-02100-f005:**
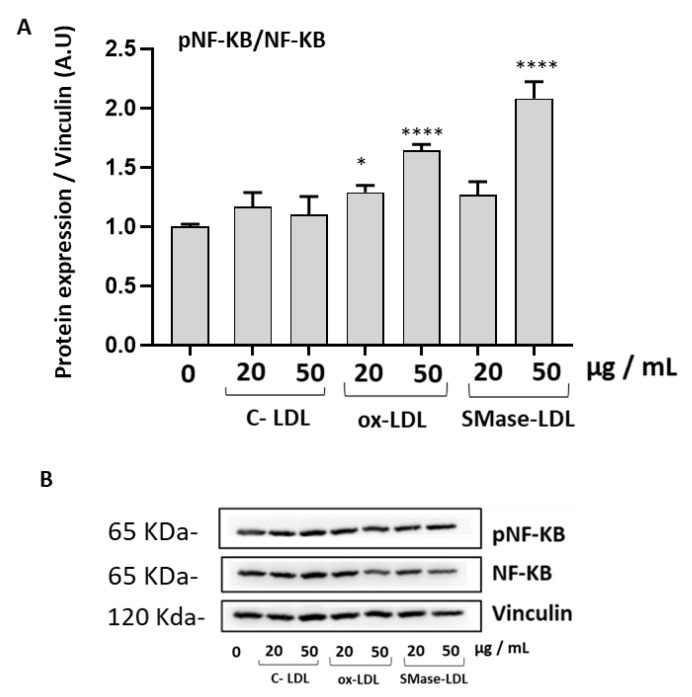
Effect of modified LDLs on levels of NF-κB -in HUVECs. (**A**) Densitometric analysis of NF-κB in HUVEC cells incubated for 24 h in the absence or in the presence of cells incubated with control LDLs (C-LDLs), oxidized LDLs (ox-LDLs) or LDLs treated with 100 mU/mL SMase (SMase-LDLs). Densitometric data are normalized on Vinculin. Results are presented as mean ± SD of 3 determinations carried out in triplicate. (* *p* < 0.05, **** *p* < 0.0001 vs. C-LDL). (**B**) Representative western blot images.

**Figure 6 molecules-28-02100-f006:**
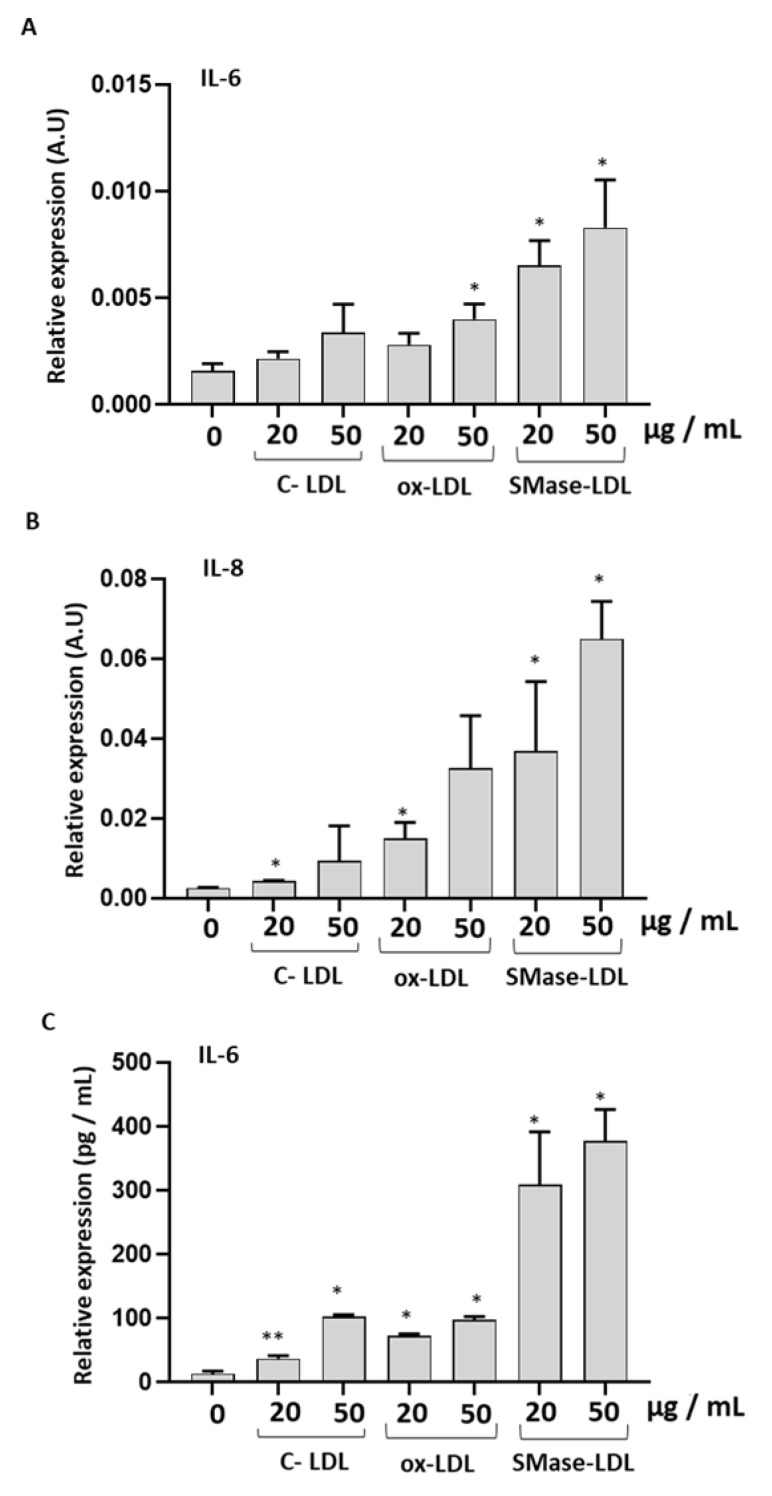
The effect of modified LDLs on levels of IL-6 and IL-8 in HUVECs. (**A**) Relative mRNA levels of IL-8, (**B**) relative mRNA levels of IL-6, and (**C**) protein levels of IL-6, in HUVECs incubated for 24 h in the absence or in the presence of control LDLs (C-LDLs), oxidized LDLs (ox-LDLs) and LDLs treated with 100 mU/mL SMase (SMase-LDLs). Densitometric data are normalized on Vinculin. Results are presented as mean ± SD of 3 determinations carried out in triplicate. (* *p* < 0.05, ** *p* < 0.001, vs. C-LDL).

**Table 1 molecules-28-02100-t001:** Physico-chemical properties of control (C-LDL) and treated LDLs in different experimental conditions. * *p* < 0.05; ** *p* < 0.001 vs. C-LDL.

	C-LDL	ox-LDL	SMase-LDL 50 mU/mL	SMase-LDL 100 mU/mL
Turbidity (absorbance a 450 nm)	0.291 ± 0.012	0.272 ± 0.017	0.545 ± 0.021	0.797 ± 0.025 **
Ceramide level (µM)	0.205 ± 0.015	8.476 ± 0.352	12.1± 0.02 *	23.843 ± 0.9 **
Fluorescenceintensity (AU)	461 ± 21	128 ± 5 *	400 ± 15	392 ± 13 *
Hyperchromicity(AU)	0.080 ± 0.005	nd	0.085± 0.010	0.100 ± 0.004 *
GP value (Laurdan)	0.52 ± 0.01	0.54 ± 0.03	0.48 ± 0.04	0.42 ± 0.02
TBARS (nmol MDA/mg protein)	0.10 ± 0.03	26.2 ± 1.3 **	nd	0.13 ± 0.03

GP, generalized polarization; TBARS, thiobarbituric acid reactive substance assay.

## Data Availability

Not applicable.

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
