# Peer review of "Effect of Sphingomyelinase-Treated LDLs on HUVECs"

_molecules, 2023, doi:10.3390/molecules28052100_

Round 1

Reviewer 1 Report

This study clearly shows effects of sphingomyelinase treated LDL on HUVECs. The results are quite impressive and methods are described adequately. Statistical interpretation shows some weaknesses because triplicates are usually not acceptable for calculation of mean and standard deviation. Even considering that cell culture experiments are complex and time consuming, n=6/7 should be targetted and/or in case of n=3 statistical methods should be applied, which are not dependent on normal distribution. (e.g. u-test instead of t-test)

However, extensive language editing is necessary because of numerous space errors especially in the materials and methods section, which must be eliminated before publication. (e.g. line 302 ...plasmawaspurchased from...; line 303 ...NaCland...; line 304 ...havebeenacquiredfrom... etc) 

Author Response

Reviewer 1

We are grateful to the Reviewers and the Editor for their constructive comments and suggestions, which helped to improve the quality of this manuscript.We are going to thoroughly discuss the points raised by the reviewers. Changes in the manuscript text are highlighted in red. We hope that the revised manuscript is suitable for publication in Molecules.

Comments and Suggestions for Authors:

This study clearly shows effects of sphingomyelinase treated LDL on HUVECs. The results are quite impressive, and methods are described adequately. Statistical interpretation shows some weaknesses because triplicates are usually not acceptable for calculation of mean and standard deviation. Even considering that cell culture experiments are complex and time consuming, n=6/7 should be targetted and/or in case of n=3 statistical methods should be applied, which are not dependent on normal distribution. (e.g. u-test instead of t-test).

We understand the Reviewer’s concerns. The choice of performing triplicates is a quite established approach for cell-based experiments. Performing additional biological replicates would be very time consuming and not feasible within the 10-day deadline that was granted for the revision. Regarding the statistical methods, we used a two-tailed paired t-test to compare conditions. In the first version of the manuscript, we did not specify that the t-test was paired, as appropriate for biological replicates. We updated the statistical analysis paragraph of the Methods. Please note that according to specialized literature, no objections to using the t-test with small sample sizes are proposed (10.7275/e4r6-dj05). Indeed, even the original Student’s paper describing the t-test (https://www.york.ac.uk/depts/maths/histstat/student.pdf) reports an example based on a sample size of 4. Moreover, use of the Wilcoxon signed-rank test for comparison of paired samples is not possible for group sizes <5, as achievement of a significant p-value would be not possible (see 10.2307/3001968 for reference).

However, extensive language editing is necessary because of numerous space errors especially in the materials and methods section, which must be eliminated before publication. (e.g. line 302 ...plasma was purchased from...; line 303 ...NaCl and...; line 304 ...havebeenacquiredfrom... etc).

We thank the Reviewer for noticing these typos, many of which were introduced due to formatting issues.

Reviewer 2 Report

In this paper, Giuliani et al. Showed by in vitro study that sphingomyelinase (SMase)-LDL has a strong proinflammatory effect compared to ox-LDL. They explored the effects of ox-LDL and SMase-treated LDL on endothelial cells, evaluating cell viability, apoptosis, pro-inflammatory status, oxidative stress and antioxidant systems. The pro-inflammatory and pro-oxidant effect of SMase is well known in the literature, so the main interesting part of this work is the comparison between SMase-LDL and ox-LDL.  However, there are some issues to better clarify and the comments for the manuscript are indicated below:

1.     How did the authors choose the concentrations for the treatments? Was a dose-response curve performed? Since the 50 mU/mL concentration was also vital for the cells (fig.1), why did the authors choose to use the higher concentration (100 mU)? Do you have any data regarding the effects of dose <50?

2.     Figure 2: the b-actin shown in the figure does not match those reported in the "original images" file. Why? Moreover, the increase of Caspase 3 expression is not well represented by western blot bands (fig 2 B). In this way, results are not supported.

3.     Only antioxidant enzymes like PON2 were assessed. Since a lot of enzymes contribute to oxidative stress, I suggest assessing the activity of other proteins, for example, NADPH oxidase. In addition, to better identify molecular mechanisms, use a PON2 inhibitor to test its involvement. Again, the increase of PON2 expression needs to be better represented by western blot bands (fig 3 B) as the results are not supported.

4.     In the methods, the authors wrote that the use of the anti-p56 antibody to evaluate NF-Kb activity, but, in this regard, the PM reported in Figure 5B is not correct. The molecular weight p65 is expected to be between 60 and 80 KDa. Please, check and correct it.

5.     The authors reported incorrectly PON2 in figure legend 5.

6.     The authors discussed the result of IL-8 but, any result was shown. I suggest adding data on IL-8 in the results or deleting it in the discussion.

7.     How do sphingomyelinases work? do they have a cell receptor? do they act through a direct or indirect mechanism?

8.     To better define the role of ox-LDL and SMase-LDL you should block the receptor on the cells or use positive and negative controls in the in vitro experiments.

9.     I suggest adding in all western blot representative bands the description of treatments.

10.  Finally, I suggest better describing the statistical analysis as the replicate number of experiments is low.

Author Response

Reviewer 2

We are grateful to the Reviewers and the Editor for their constructive comments and suggestions, which helped to improve the quality of this manuscript.We are going to thoroughly discuss the points raised by the reviewers. Changes in the manuscript text are highlighted in red. We hope that the revised manuscript is suitable for publication in Molecules.

Comments and Suggestions for Authors:

In this paper, Giuliani et al. Showed by in vitro study that sphingomyelinase (SMase)-LDL has a strong proinflammatory effect compared to ox-LDL. They explored the effects of ox-LDL and SMase-treated LDL on endothelial cells, evaluating cell viability, apoptosis, pro-inflammatory status, oxidative stress and antioxidant systems. The pro-inflammatory and pro-oxidant effect of SMaseis well known in the literature, so the main interesting part of this work is the comparison between SMase-LDL and ox-LDL.  However, there are some issues to better clarify and the comments for the manuscript are indicated below:

  1. How did the authors choose the concentrations for the treatments? Was a dose-response curve performed? Since the 50 mU/mL concentration was also vital for the cells (fig.1), why did the authors choose to use the higher concentration (100 mU)? Do you have any data regarding the effects of dose <50?

The treatment of LDL with SMase to obtain aggregation of LDL and lipoprotein particles with a higher content of ceramide has been previously used by other authors as experimental model of a physiologically important atherogenic lipoprotein  using different concentrations of SMase (Sneck, M.; Nguyen, S.D.; Pihlajamaa, T.; Yohannes, G.; Riekkola, M.L.; Milne, R.; Kovanen, P.T.; Oorni, K. Conformational changes of apoB-100 in SMase-modified LDL mediate formation of large aggregates at acidic pH. J Lipid Res 2012, 53, 1832-1839; Xu, X.X.; Tabas, I. Sphingomyelinase enhances low density lipoprotein uptake and ability to induce cholesteryl ester accumulation in macrophages. J Biol Chem 1991, 266, 24849-24858; Oesvstang eta al. Modification of LDL with human secretory phospholipase A2 or sphingomyelinase promotes its arachidonic acid-releasing propensity. J Lip Res 2004, Pages 831-838).  In fact both LDL and sphingomyelinase are present in atherosclerotic lesions and there is a possibility that sphingomyelinase interact with LDL and participate to molecular mechanisms of atherosclerosis. In detail, a growing interest has been devoted to the physio-pathological role of ceramide in the molecular mechanisms of atherosclerosis. In a preliminary phase of our study we evaluated physico-chemical properties of LDL incubated with SMase  at different levels (10mU, 25 mU, 50mU and 100 mU/ml) and we used different times of incubation at 37°C. No significant changes of turbidity and other biochemical parameters were observed at concentrations of SMase lower than 50 mU/ml.. Based on results shown in table 1, we chose to use the higher concentration of SMase in incubations with HUVEC cells. In fact In our experimental conditions treatment with SMase 100 mU/ml induced a higher formation of ceramide, and stronger modifications of LDL compared to 50 mU/ml, without negatively affecting cell viability.

  1. Figure 2: the b-actin shown in the figure does not match those reported in the "original images" file. Why? Moreover, the increase of Caspase 3 expression is not well represented by western blot bands (fig 2 B). In this way, results are not supported.

We agree with the Reviewer and apologize for the misunderstanding. We have now corrected the figure 2 with the right images.

  1. Only antioxidant enzymes like PON2 were assessed. Since a lot of enzymes contribute to oxidative stress, I suggest assessing the activity of other proteins, for example, NADPH oxidase. In addition, to better identify molecular mechanisms, use a PON2 inhibitor to test its involvement. Again, the increase of PON2 expression needs to be better represented by western blot bands (fig 3 B) as the results are not supported.

We agree with the Reviewer. As suggested, we added SOD2 expression in figure 4 and in the main text.We also thank the Reviewer for the insightful suggestion of investigating the involvement of PON2 in endothelial response to modified LDL, which will be deepened in our future studies.

Regarding the PON2 WB, we are confident that the blot reported in Fig. 4B is enough representative of the results obtained across replicates. To address the Reviewer’s concerns, here we provide the WB of another replicate.

  1. In the methods, the authors wrote that the use of the anti-p65 antibody to evaluate NF-Kb activity, but, in this regard, the PM reported in Figure 5B is not correct. The molecular weight p65 is expected to be between 60 and 80 KDa. Please, check and correct it.

We apologise for the mistake.We updated the figure with the actual molecular weights.

  1. The authors reported incorrectly PON2 in figure legend 5.

Thank you. Mended.

  1. The authors discussed the result of IL-8 but, any result was shown. I suggest adding data on IL-8 in the results or deleting it in the discussion.

If we understood well, probably the Reviewer may not have noticed that results of IL-8 are introduced in lines 162-164 of the original Manuscript and displayed in Figure 6A.

  1. How do sphingomyelinases work? do they have a cell receptor? do they act through a direct or indirect mechanism?

As aforementioned, the treatment of LDL with SMase  in vitro has been previously used to obtain a useful experimental model of a physiologically important atherogenic lipoprotein. Some sentences have been added to describe better the effect of  SMase on lipid composition of LDL. Moreover we revised the manuscript and new references have been added concerning the study of the interactions between LDL treated with sphingomyelinase and different cell models. Aggregation of LDL occurs after Smase treatment and we confirmed that in our experimental conditions aggregation realizes at the end of incubation between LDL and SMase. Literature data demonstrate that aggregated LDL can be internalized by a LDL receptor or by other mechanisms involving plasma membrane invaginations. Previous studies have demonstrated  uptake of SMase-LDL mediated by LDL receptors (Xu, X.X.; Tabas, I. Sphingomyelinase enhances low density lipoprotein uptake and ability to induce cholesteryl ester accumulation in macrophages. J Biol Chem 1991, 266, 24849-24858.) in macrophages. In addition, using cytochalasin D during incubations, authors demonstrated that endocytosis, not phagocytosis, was involved in internalization of sphingomyelinase-treated LDL. Boyanovsky et al ( 2003) have confirmed that uptake of ceramide-enriched LDL by human microvascular endothelial cells in a receptor-mediated fashion. (Gupta, A.K.; Rudney, H. Sphingomyelinase treatment of low density lipoprotein and cultured cells results in enhanced processing of LDL which can be modulated by sphingomyelin. J Lipid Res 1992, 33, 1741-1752. Boyanovsky, B.; Karakashian, A.; King, K.; Giltiay, N.; Nikolova-Karakashian, M. Uptake and metabolism of low density lipoproteins with elevated ceramide content by human microvascular endothelial cells: implications for the regulation of apoptosis. J Biol Chem 2003, 278, 26992-26999)

See revised discussion

  1. To better define the role of ox-LDL and SMase-LDL you should block the receptor on the cells or use positive and negative controls in the in vitro experiments.

In methods section of the revised manuscript, we described better the experimental models. In all experiments, we compared biochemical parameters in cells incubated with buffer without lipoproteins (lipoprotein –free, negative controls) and in  cells incubated with the presence of LDL treated in different experimental conditions: LDL incubated for 24 h at 37°C without enzyme (control LDL), LDL incubated with SMase (SMase-LDL) or LDL incubated with  copper ions (Ox-LDL).

The effect of Ox-LDL has been widely investigated and their cytotoxic roles have been previously demonstrated using different cell models. The topic has been also reviewed. The effect of SMase-treated LDL on HUVEC has not been previously studied. We decided to study the effect of LDL treated with SMase in our experimental conditions and we compared the results obtained using Ox-LDL as reference of atherogenic lipoprotein.

  1. I suggest adding in all western blot representative bands the description of treatments.We agree with the Reviewer ,all figures have been updated.
  2. Finally, I suggest better describing the statistical analysis as the replicate number of experiments is low.

We understand the Reviewer’s concern. All the experiments were performed on three independent biological replicates, and differences among conditions were assessed using paired t-tests. The statistical analysis paragraph has been updated.

Round 2

Reviewer 2 Report

Thanks to the authors for evaluating the suggestions and answering carefully. I have no further comments.